# A Narrative Review about Prosocial and Antisocial Behavior in Childhood: The Relationship with Shame and Moral Development

**DOI:** 10.3390/children9101556

**Published:** 2022-10-14

**Authors:** Susanna Maggi, Valerio Zaccaria, Maria Breda, Maria Romani, Franca Aceti, Nicoletta Giacchetti, Ignazio Ardizzone, Carla Sogos

**Affiliations:** Department of Human Neurosciences, Sapienza University, viale dell’Università 30, 00185 Rome, Italy

**Keywords:** shame, childhood, moral development, prosocial behavior

## Abstract

We conducted a literature review aimed at identifying the origins of shame as well as its effects on moral development, especially in terms of behavioral outcomes, and we reflected on the practical implications of our findings. We explored the role of shame in moral development through cultural differences and parental influences, collecting evidence of psychopathological consequences of primary moral emotion dysregulation. These studies showed a dichotomous feature of shame, as a prosocial behavior enhancer in morally relevant situations and, simultaneously, a risk factor for aggressive and antisocial behaviors on other occasions. Dysregulated shame leads to maladaptive interpersonal behaviors, which could evolve towards psychopathological paths. Therefore, an integrated intervention is recommended in children with emotional/behavioral problems.

## 1. Introduction

Moral concerns regulate our daily interactions with others, therefore heavily influencing both our reasoning and our behavior. Nevertheless, there are still many unresolved questions about morality and, most of all, moral development during childhood, such as: do moral judgements rely on automatic, affective reactions or do they come from reasoning about internalized moral principles? [1] There is growing evidence that both cognition and emotion are fundamental in moral development as well [2], since they cooperate in prompting the rules and the internalization of moral values, through both cognitive mechanisms and the so-called moral emotions, such as guilt and shame.

Guilt and shame, although sharing common characteristics, are significantly distinct [3], and therefore, their investigation need to aim at identifying the unique correlates they rely on.

Indeed, Bastin and colleagues, in a study published in 2021, provided evidence that guilt and shame may even rely on distinct brain regions. In their research, 36 healthy females (mean age 18.8 ± 1.9 years) were given 44 social moral dilemmas and then underwent an MRI, showing that shame was associated with decreased activation of the superior temporal gyrus and precentral gyrus in addition to decreased activation of the middle frontal gyrus during reflection, while guilt related mainly to precuneus and putamen decreased activity [4].

So far, professionals focused their attention mainly on exploring the impact that guilt has in terms of behavioral and psychological outcomes in childhood, but lately, the interest in investigating the role of shame is blooming as well.

Shame is a self-conscious emotion, which means it involves some form of self-evaluation or self-reflection [5]; in contrast to basic emotions such as joy and sadness, which emerge earlier in life, self-conscious emotions are described as “secondary” because they surface later in life, when children develop key cognitive abilities which allow them to “recognize” themselves: this is a necessary prerequisite to experience a self-conscious emotion [6,7].

However, this is not the whole story: given that self-conscious emotions are so strongly related to the self, they also regulate intimately our relationships with others and, therefore, include a very social dimension, typically arising in social situations [6,8,9,10,11,12].

This makes it easier to comprehend why self-conscious emotions can prompt important and different interpersonal behaviors [8,11,13]. This is particularly true since shame is described as “usually dependent on the public exposure of one’s frailty or failing” [14], showing itself mostly when feeling exposed: this stands true even if there’s no real observer witnessing one’s shortcoming because shame is often experienced while imagining about how one’s defective self would appear to others [5].

Shame is also a complex emotion which can be looked at as a two-sided one: arising from moral situations, such as violating a moral standard (“moral shame”) or springing out from non-moral situations, such as a personal failure in front of an audience (“non-moral shame”) [5].

In both situations, shame represents a painful emotion that implies a devaluation of the entire self and dispositional inadequacy, often making the individual feel as if he/she was “naked” and exposed to others’ judgement.

Therefore, it is not a big surprise that self-conscious emotions, such as shame, have been related to relevant behavioral consequences in people, also through the mediating effect of empathy: it has been shown that people induced to feel shame display less empathy [15] and that shame-proneness relates to an impaired capacity for other-oriented empathy.

Consistently with these findings, numerous empirical studies of both children and adults show that shame-proneness relates to proneness to feelings of anger and hostility too [13,16,17,18]: all of these mechanisms combined together end up creating distance, separation and eventually aggressive behaviors toward others, deeply undermining an individual’s capacity to build healthy relationships and positive social interactions with other people.

This seems to happen as a direct consequence of an intense hostility initially felt toward the self: later, in a defensive attempt to protect the self, this hostility can be redirected outward, shifting the blame elsewhere and finally accounting for the “shame-rage spiral” described by Scheff [19,20] and Retzinger [21].

Since shame seems to have such a heavy impact on one’s behavior and ability to socially interact with others, we aimed to identify the main factors influencing shame’s development and intensity during childhood; our findings allowed us to reflect upon consequences of shame dysregulation and potential relations between shame’s management and children’s behavior, especially in terms of prosocial or antisocial tendencies; additionally, we related our findings to consequences upon children’s behavior, especially in terms of proneness to prosociality or antisociality. Indeed, in contrast with previous theories, which claimed that emotions were just disruptive bias to moral thought, recently, philosophers and psychologists have agreed that higher-order emotions such as empathy, guilt, and shame play a crucial role in the development of morality [22].

Piaget thought morality is neither born with the individual nor just imposed from the collectivity, but it is rather built up on social interactions [23], a vision shared by the more recent social domain theory [24,25,26], the idea of which is that morality is grounded in social interactions: since it has been observed that shame, as we have briefly described, deeply influences social interactions, we hypothesized that it may condition moral development through childhood.

Relying on this hypothesis, we aimed to identify and describe origins of shame, as well as its effects on moral development and behavioral outcomes, reflecting on the potential practical implications of our findings.

## 2. Materials and Methods

### 2.1. Search Strategy

The present study was carried out through a literature review aimed to identify qualitative studies that described the impact that shame has on moral development of the child. Five databases (PubMed, PsycInfo, ISI Web of Science, Cinahl, and the Cochrane Database of Systematic Reviews) were queried using the following terms: *(child OR children OR childhood OR kid* OR pediatric* OR paediatric*) AND (shame OR self-blame OR self blame OR self-criticism OR self criticism) AND (moral development OR moral sense OR moral self OR prosocial behavior* OR antisocial behavior*)*. No filters were applied for study design or date of publication. All the available studies published up to July 2022 were retrieved. Any further relevant data were also searched for by reviewing the references of considered studies. The titles and abstracts of retrieved studies were initially selected by three independent reviewers (M.B., S.M., and V.Z.) based on their pertinence and relevance to the topic of the review. Discrepancies were resolved by discussion among the reviewers. The full texts of the selected studies were obtained and read, checking eligibility with inclusion criteria.

### 2.2. Inclusion Criteria

In the review, we included studies that met the following criteria: studies that were (a) peer-reviewed original research papers; (b) published and available in English; and (c) specifically focused on self-conscious emotions, specifically shame, and their relationship with moral development in children aged up to 12 years old. We did not include in the review (a) studies focused exclusively on adolescents and adults (i.e., >12 years old); (b) papers focused on the moral development of the child but not on the role of shame in the process; and (c) different study designs such as editorials, abstracts, posters, conference proceedings, and doctoral theses.

### 2.3. Data Synthesis

Data were extracted from each included paper, evaluating the design of the study, the characteristics of the participants, and the tools used to measure or describe shame within the sample, if applicable. A narrative synthesis of studies was realized in order to clarify the heterogeneity of research methodologies being employed across studies.

## 3. Results

### 3.1. Search Results

Our search produced a total of 258 records, which were initially selected on the basis of pertinence to the topic of the study and then with application of the inclusion/exclusion criteria mentioned above. This led to 58 included studies, which have been sorted by the main topic: shame and cultural differences, shame and parental influences, and the psychopathological consequences of proneness to shame.

### 3.2. Moral Development through Cultural Differences: The Role of Shame

Society gives an important contribution to moral development, shaping personal behaviors and values through the constant influence of other members of the community. Thus, cultural differences play a crucial role in the construction of a moral identity in the growing child as well as in the expression of his/her moral emotions. Culture, as the totality of the characteristics and knowledge of a particular group of people, including its language and social habits, affects the way a certain society looks at moral emotions.

For example, shame, though being universally recognized and experienced [27,28], as well as guilt, are profoundly influenced by how a culture considers the nature of the self and, consequently, the experience of this self-conscious emotion. In 1946, the anthropologist Ruth Benedict coined the expressions “guilt culture” and “shame culture” [29] as the result of her studies of two different societies: the North American and the Japanese. A guilt culture, such as the North American and other Western societies, emphasizes individual conscience, encouraging self-control in the face of temptation and reinforcing guilt for a certain condemned behavior in fear of a punishment. Consequently, communities where guilt is predominant have members with a stronger conscience, and they are generally referred to as individualistic cultures. On the other hand, a shame culture, such as many Eastern societies, gives much more importance to shame and respect, emphasizing the opinion that others might have about what the individual does and thinks. In these cultures, generally defined as collectivistic, moral guidance is often delegated to the extended family and the whole community, which take a disciplinary role much more than in guilt-based cultures. Therefore, these individuals will develop a weaker conscience. Grinder and colleagues [30] explored the effects which shame cultures and guilt cultures have on conscience development, studying in particular two different populations (Samoan and American Caucasian children). The research was conducted in Hawaii, where previous studies showed that Samoans have built their community entirely upon shame [31]. Overall, 34 children (15 American Caucasians and 19 Samoans) were enrolled from a rural, public school. The authors assessed resistance to temptation through a realistic game situation involving a “ray-gun” shooting and a target box, with a scoring system and the consequent reward on the basis of participants’ accuracy, observing how many children falsified their scores in order to earn the reward. Furthermore, guilt was assessed by administering five incomplete stories revolving around the violation of socially expected behavior patterns (honesty, trustworthiness, or self-control), and evaluating remorse, confession, and restitution as three main dimensions of guilt itself. The results illustrated, as expected, that a higher proportion of Samoans than American Caucasian children showed a weaker conscience: none of the 19 Samoans resisted temptation, while seven of the 15 American Caucasians did. Nevertheless, Samoans had a moderately strong disposition toward guilt, suggesting rudimentarily, internalized aspects of conscience.

In 2012, Furukawa and colleagues [32] conducted a study aimed at exploring cross-cultural differences in self-conscious emotions and their psychosocial correlates among children residing in Japan, Korea, and the US. Overall, 144 Japanese children aged 8–9 years, 180 Korean children aged 10–11 years, and 688 US children aged 9–11 years were enrolled. The authors used the Test of Self-Conscious Affect for Children (TOSCA-C) to measure children’s proneness to feel shame, guilt, and pride, presenting brief scenarios with common situations, and asking the child to specify on a 5-point scale the likelihood of reacting in the same way as shown in the illustration. As expected, Japanese children scored higher on shame than Korean and US children, while females always scored higher on shame across the three samples. Furthermore, Korean children scored higher on guilt than US and Japanese children, with the latter group in an intermediate position. Gender did not have a significant effect in this case. Moreover, possible correlations of self-conscious emotions with externalizing dimensions (externalization of blame, anger, and aggressive behaviors) were investigated. Similar results were observed for shame among Japanese, Korean, and US children; in US and Korean samples, shame was positively correlated with those externalizing dimensions, while it was positively associated with anger and externalization of blame in the Japanese group, but unrelated to aggressive behaviors.

### 3.3. Moral Development and Shame through Parental Influences

How do parental’s disciplining techniques impact children and adolescents’ prosocial and moral values and behaviors? The crucial role parents play in the moral development of their children was clearly pinpointed by Freud’s concept of superego as coming from a progressive process of internalization of parents’ values and prohibitions. However, Freudian theory gives little attention to parents’ active role in children’s disciplining [33,34], since their teaching styles can meaningfully affect children’s moral emotions, possibly leading to higher levels of shame, which is itself linked to internalizing and externalizing problems.

Inductive discipline, love withdrawal, and power assertion are different discipline techniques in which the relationship with positive/negative psychological outcomes has been analyzed and is reported here.

Induction refers to a specific teaching style that can be summarized as “reasoning with the child”, pointing out to him/her the consequences of the child’s transgression for the victim; by inducing empathy-based guilt in the child, this discipline technique can be described as “other oriented” and it is thought to help children internalizing parental moral values [35,36].

Hoffman [36,37] thought inductive discipline was more effective than love withdrawal or power assertion in prompting prosocial behaviors because of the aforementioned empathy-based guilt: indeed, children and adolescents whose parents use inductive discipline display more guilt and reparative behaviors.

In contrast, love withdrawal and power assertion, together with shaming and conditional regard, are disciplinary strategies which children perceive as forms of psychological control because they focus on social expectations and criticism toward the self, generating shame and anxiety for love loss and eventually dysregulating children’s levels of shame. In particular, love withdrawal implicates communicating to the child that he/she must be obedient or successful to deserve love and affection [38], since it is a discipline technique consisting of retaining affection when a child misbehaves or fails; not surprisingly, it is considered psychological maltreatment when used excessively [39].

Similarly, power assertion has been linked to increased levels of shame in children, along with impaired self-regulation, increased aggression and conduct problems [40,41,42,43]: it consists of physical punishment, loss of privileges, psychological aggression, and penalty tasks [44]. In a study conducted in 2003 by Kochanska [45], mother–child dyads were observed from 14 to 73 months in different contexts: “do” and don’t” discipline contexts and mother–child discourse contexts (parent–child conversations about the child’s past conduct, employed here to investigate the moral cognition domain). Researchers aimed at finding linkages between a mother’s power assertive style and moral cognition/moral conduct and antisocial conduct in children. The results showed less internalized moral conduct in the 5th and 6th years in children who had experienced more maternal power assertion in the prohibition contexts. Additionally, higher maternal power assertion in the discipline contexts between 14 and 45 months added 15% of unique explained variance in the ratings of children’s antisocial conduct by mothers and teachers at 73 months. To explain behavioral outcomes, it was proposed that maternal power assertion may impair children’s future moral behavior by leading them to externally attribute their anger and resentment, also perceiving this disciplinary strategy as a threat to autonomy; in regard to the moral cognition domain, indeed, researchers hypothesized that power assertion may lead children to develop severe anxiety and to “self-focus” in an attempt to defend themselves, reducing their attention to others’ needs and feelings and impairing their moral judgments.

In regards of shaming, it has been shown that parents can actually approach their children’s shame management in an adaptive or a maladaptive way, leading, respectively, to reintegrative or disintegrative shame. Parents’ stigmatizing shaming correlates with shame displacement in the child, which means unacknowledged shame, blaming, and anger toward others: this relates to high proneness to bullying.

In contrast to disintegrative shame, instead, if shame is approached in a “reintegrative way”, which means parents are able to provide a warm and comfortable relationship, combined with respect, children develop a better self-emotional regulation and score lower on bullying [46]. Indeed, Reintegrative Shaming Theory states that individuals’ ability to manage shame and, subsequently, their behavior, depends on their bonding to those who offer shaming as well as the type of shaming they receive. Shaming can be described as “the social process of expressing disapproval of the wrongdoing” [47], and it is thought to have a ‘‘conscience building effect’’: this is why it can be used as a powerful regulatory practice. Indeed, the way shaming is delivered is fundamental in determining if the perpetrator will manage possible feelings of shame in adaptive (e.g., making amends) or maladaptive ways: in this case, “disintegrative shaming” is equivalent to stigmatization and it loses any regulatory power. Therefore, shaming can and must be offered in a reintegrative way, communicating both disapproval of the transgression and respect.

Reinforcing the theory that parental teaching style can profoundly alter the way children experience and manage shame, it has been observed that children from highly restrictive parents display greater levels of shame and actually transgress more often than children from less punitive parents. Janoff-Bulman [48] hypothesized that this could be due to a kind of overregulation, which means children do internalize moral values but are taught to think more about “immorality” and the “do not’s” rather than the “do’s”, since their parents’ discipline involves proscription more than prescription.

Giving support to this vision, a study involving girls aged 3 to 5 found that maternal and paternal authoritarian parenting (demanding, harsh, and unresponsive parenting) predicted girls’ shame responses [49]. Mills and colleagues [50] also found that maternal shaming predicted shame responses in children from preschool to school ages.

As previously suggested, it is interesting to see how these correlations between parental styles and psychological outcomes in children get mediated by whether the child perceives his or her parents as psychologically controlling or not. Children begin to detect psychological control at about 8 years of age. This perception influences internalization of rules and values and is itself influenced by three parameters: which domain the blamed behavior belongs to, the transgression’s victim as highlighted by parents, and whether the specific behavior or the whole individual is being criticized.

By talking about “which domain” parents may blame, we refer to the social domain theory, which states that individuals’ social knowledge develops in three domains: the moral, societal, and psychological. Moral (e.g., assaulting or stealing) and societal behaviors (e.g., eating with appropriate utensils) can be legitimately regulated by others because they concern welfare or social order, and therefore, they can be targeted more easily as right or wrong. In contrast, personal issues (psychological domain) concern preferences and choices uniquely affecting the actor (e.g., friendship, hobbies, etc.) and, therefore, cannot be regulated by others [51,52]. It has been observed that children more easily accept being corrected by their parents if the action they criticize belongs to the moral or social domain rather than to the psychological domain. In other words, children tend to perceive it as fair if they are being criticized for harming another individual or disrupting social order, while they may see their parents as malicious and psychologically controlling if they argue about their friends or how they like to dress up, causing them to reject more easily what they are being taught.

The highlighted “victim of the transgression” represents another fundamental parameter in determining whether children will perceive their parents as psychologically controlling: in other words, parents can either address the effective consequences of the transgression on the direct victim of the action (e.g., another child) or they may highlight the negative emotions (such as discomfort, anger, or distress) they are experiencing because of the child’s transgression [53]. In the first case, children’s understanding of the negative effects of their behavior becomes stimulated, and they are more likely to function effectively within society’s moral standards [36]; in contrast, if parents tend to induce guilt over indirect harm to themselves, they impair children’s moral understanding: also, children will end up feeling over-responsible for their parents’ distress, leading to maladaptive outcomes [54].

Lastly, it has been observed that children tend to experience maladaptive guilt and shame if, in consequence of a transgression, parents tend to blame the whole person, the child themself, rather than the specific action/behavior: in this case, children end up feeling as if they are being criticized as individuals, for what they are, rather than for their actions.

Summing up, we can then say that children are able to detect if their parents are not acting with the child’s best interests at heart, and this is the main mediating feature ultimately leading to altered internalization of rules and values, shame displacement, and bullying.

Indeed, given that shame is a self-conscious emotion, it is profoundly linked to how the child sees himself/herself, and this feedback is provided primarily by caregivers. This makes it clear why parental discipline techniques are strongly linked to children’s socioemotional functioning.

In addition to parental teaching styles, other variables have been studied in order to define how parents may impact their offspring’s moral development; in this regard, we found a systematic review [55], the aim of which was to identify any relation between attachment styles and the development of moral emotions: specifically, the results showed that a secure attachment at 14 years of age related to increased empathic support during observed interactions with friends across ages 16 to 18, while less secure teens developed these skills slower [56]. Furthermore, empathic sensitivity was highest in those youths with low attachment anxiety, while high attachment anxiety related to lower levels of empathic sensitivity [57].

Parent attachment did not relate directly to social behavior; it was rather mediated by aspects of emotional competence such as empathy itself but also emotional awareness and positive expressiveness [58].

### 3.4. The Psychopathological Consequences of Proneness to Shame

Given the role of proneness to shame in prompting internalizing problems and depressive symptoms, which in turn can influence social functioning, and given the role of shame as a moral emotion, we aimed to investigate if variations in the level of shame may condition a child’s moral development and/or have behavioral consequences. What happens to a child’s moral development when one primary moral emotion such as shame is dysregulated and/or shame management is not fulfilled?

Shame is related to poor interpersonal adjustment [17,59] and has been linked to social anxiety and phobia, as well as to an increase in depressive mood [60,61]. This is consistent with findings relating a thinner posterior cingulate cortex (PCC) with shame and with depression [62,63] and social anxiety disorder [64]; indeed, a study published in 2016, [65] where sixty participants (aged 15–25) completed the Experience of Shame Scale (ESS) [66] and underwent an MRI, showed that higher levels of shame-proneness were associated with thinner right PCC. In the same study, higher levels of shame-proneness also related to smaller right amygdala volumes, suggesting a potential link between the experience of shame and amygdala’s role in processing aversive emotion and social threat [67]. This may help understand how significant levels of shame can both lead to hiding the self or withdrawal [68,69,70] and, sometimes, relate to greater externalizing tendencies to the extent that it elicits feelings of the wounded self, trying to defend itself [69,71]. Some studies have actually shown that proneness to shame is positively correlated with aggression and negatively correlated with prosocial behavior in both children and adults, while the exact opposite occurs in case of proneness to guilt [16,72].

In regard to antisocial behavior, Ortiz Barón et al. [73] conducted a study with 351 children aged 10–14 confirming the negative association between shame and prosocial behavior and the positive one with antisocial behavior, although to a lesser extent than the relation between empathy and guilt: high and low empathy levels were not found to affect antisocial behavior when children had high guilt levels, while in children with low guilt levels, low empathy levels predicted significantly higher antisocial behavior scores. Furthermore, this study had some limitations, including the absence of a distinction between moral and non-moral shame. Ignoring this distinction may be one of the reasons why some studies link shame to aggressive behaviors and moral disengagement, while in contrast, other studies have found this emotion to have a functional value in fostering prosocial behavior and controlling antisocial behavior [74]. Indeed, aggressive behaviors are evident moral transgressions, and the literature on traditional bullying and aggression suggests that moral emotions are important regulators of harmful behavior, as they are closely connected with one’s sense of responsibility toward another [75,76]. In this vision, moral shame, along with guilt, indicates a recognition of the harmful consequences inflicted on the victim [76], and in fact, bullies have been found to experience less shame and guilt [75], feeling greater indifference to victims’ suffering [77]. From this perspective, moral shame appears to implicate acceptance of personal responsibility, refraining from further wrongdoing, withdrawn behavior [78,79], and making amends when a moral standard is violated (e.g., harming a peer [74]).

Other authors have obtained coherent results in confirming the role of shame as a moral emotion which fosters internalization and acceptance of moral standards, since it has been found that shame inhibits antisocial behavior [78], while, in contrast, the tendency to rarely experience shame and guilt is associated with maladaptive social outcomes, such as aggression, bullying, and delinquent behavior [76,79,80].

Therefore, shame seems to come with a high level of internalization of moral values, likely leading a child to better self-control and moral behavior, while moral disengagement, in contrast, can represent a powerful mechanism able to fuel aggressive behavior [81]. However, to fully understand shame’s effects on moral development and behavior, it seems to be fundamental not to forget about the double-sided nature of shame as noted above. While guilt is associated with moral transgression, shame appears to be linked both to moral contexts and non-moral ones, such as experiences of inferiority, incompetence, or derision [82]. In particular, Olthof [78] highlighted that some situations elicit only shame and are characterized by non-moral actions (“shame-only”, SO situations; e.g., falling asleep in the classroom during lessons), while some other situations are able to elicit both shame and guilt since they are characterized by moral transgressions occurring when inept behaviors cause harm to someone (“shame-and-guilt”, SAG situations; e.g., breaking a friend’s toy). Menesini et al. [74] conducted a study in Italy, dividing into three groups 121 children aged 9–11, using peer nominations: there were then bullies, victims, and prosocial children. This study found that prosocial children experience high levels of shame and guilt in SAG situations, which are designed to elicit morality. In SAG situations, shame and guilt appear to be closely linked to the development of social skills, conscience, moral reasoning, avoidance of transgression, and a willing to repair (guilt) or to internally sanction (shame) the self [83,84]. As has been observed recently in the literature, prosocial children have a tendency to act with concern towards others through empathy and sympathy, and this could explain why they experience more feelings of guilt and shame in a moral situation [36,85]. The lower levels of shame and guilt experienced by bullies confirm and extend the results of Menesini and colleagues [76], who described bullies as prone to feelings of indifference and pride when harming someone. In addition, bullies were found to feel significantly less guilty and ashamed than prosocial children, especially in SAG situations but not in SO situations, giving strength to the hypothesis that shame can have a differential impact on the moral development and the behavioral responses of a child, either in a positive or negative way, depending on which situations (moral or non-moral) shame was elicited by.

There may be other factors influencing and mediating the effect of shame on moral development, and a study conducted by Roos et al. [86] provided an interesting point of view on this topic, involving emotion regulation capacities and negative emotionality. Results showed that guilt and shame related to lower levels of aggression for children with poor emotion regulation (or high negative emotionality). For children with effective emotion regulation (or low negative emotionality), shame and externalization of blame, which is a moral disengaging mechanism, were associated with higher levels of aggression. In this regard, it was observed that children prone to feel guilty are less likely, while shame-prone children are more likely, to externalize blame, giving support to the idea that externalizing blame may help to regulate and downsize shame, therefore reducing painful feelings of threat to the self, elicited by shame itself [71,72,87]. The hypothesis then is that effective emotion regulation may involve a “dark side”, functioning itself as a disengaging mechanism from the inhibiting role of guilt and shame, to the extent that children can act aggressively in response to shame instead of feeling responsible and therefore avoiding antisocial behaviors.

Previous studies have showed that children high in moral disengagement also show high levels of aggressive behaviors [75,80,88,89,90]. In regard to regulation capacities, children who strive to regulate their emotions, leaning toward the tendency to experience negative emotions, such as anger and sadness, are more likely to engage in problem behaviors [91,92,93]. Specifically, previous studies have suggested that poor emotion management relates to aggression [94]. Roos [86] conducted a study with 307 Finnish participants (M_age_ = 11.9 years) and assessed their proneness to guilt, shame, and externalization of blame, emotion regulation/negative emotionality, and aggressive behavior. The results indicated that, as emotion regulation increased, the inhibiting effects of guilt and shame on aggression were weakened, eliminated, or transformed (in the case of shame) into an aggression-supporting effect. This means that guilt and shame inhibited aggression in children who scored low in emotion regulation, while proneness to feel shame related to aggression in children who scored high in emotion regulation. Additionally, at high levels of negative emotionality, proneness to feel guilt and shame and to externalize blame all inhibited aggressive behavior. However, as levels of negative emotionality diminished, these inhibiting effects were weakened, eliminated, or transformed (in the cases of shame and externalization of blame) into being supportive of aggressive behavior. Thus, for children relatively low in negative emotionality, the aggression-supporting effects of shame and externalization of blame became relevant. Shame is, thus, associated with greater aggression only when emotions are effectively managed, which would also clarify the inconsistent shame–aggression associations reported in the literature [16,76].

## 4. Discussion

The findings we obtained tried to investigate social, familiar, and cultural origins of shame and how it affects children’s behavior, also through the mediating effect of empathy and moral values’ introjection. As a matter of fact, shame has a great complexity, which involves both aspects of prosocial adjustments and perilous global and negative evaluations of the self, pushing individuals on the one hand to behave and on the other to reject social boundaries and empathetic responses. Studies have shown a double-sided nature of this emotion, providing a possible explanation of the dichotomous features of the evoked responses: in morally relevant situations, shame appears to be a prosocial behavior enhancer, whereas it seems to be a risk factor for aggressive, antisocial behaviors and/or withdrawal on different occasions. By investigating parental discipline techniques, we talked about shaming as a risk factor for detrimental effects onto children’s psychological well-being, possibly leading to counterintuitive higher rates of aggressive and immoral behaviors in children whose parents tend to widely use shaming as a teaching style: this may happen because children perceive it as a reiterative aggression to the self, causing them to feel worthless, angry, or defective, as they grow up. Eventually, children can develop either internalizing problems, when the self-devaluation prevails, or externalizing problems, when they redirect their negative feelings to the outside.

Despite these consistent findings, it is been theorized that shame may play also a very functional role, regulating social interactions: therefore, it is been hypothesized that shame can lead to maladaptive interpersonal behavior, ultimately leading to psychopathological outcomes [95], when dysregulated. Giving support to this theory, as we reported before, the Reintegrative Shaming Theory postulates that parents can either approach their children’s shame in an adaptive or maladaptive way: therefore, psychopathological outcomes and behavioral disturbances may relate to how shame is managed and not to shame itself. However, since our findings were inconclusive, we hope future research helps to clarify shame’s role in prompting or impairing social interactions.

In order to reach the ambitious target of understanding the nature and consequences of shame, we conducted a literature search, which was not exempt from limitations. First, the studies we collected show a significant heterogeneity concerning sociodemographic characteristics, such as age and gender, as well as the aims of the studies, and, therefore, results and outcomes. Due to this lack of homogeneity, it was not possible to come to a univocal assessment of the role of shame in prompting prosocial/antisocial behaviors and moral values’ internalization. It is possible that variables we did not consider may influence the mediating effect of shame on a child’s morality and behavior, since the aforementioned factors may add variability to them and, therefore, involve different interpretations and meanings, depending on the context. A second limitation is related to the fact that shame and guilt are often discussed together with the intent to assess the morally relevant implications of the one as compared to the other, causing our aim to specifically analyze how shame influences behavior and morality to be only partially fulfilled. Finally, we mainly analyzed the effects of shame through childhood in terms of behavioral responses; this may cause a lack of information about inner perceptions and internalization of rules and values, leading to a misinterpretation, or partial understanding of the results.

Therefore, in our view, future research should focus on deepening our knowledge about how shame could alter our children’s moral development not only by observing the behavioral responses but also by analyzing specifically how children’s management of shame can weigh on the internalization of moral principles, regardless of behavioral tendencies. Indeed, cognitive behavioral interventions on youths (and their parents) with externalizing and internalizing problems are particularly focused on learning appropriate responses to primary emotions such as fear, sadness, and anger. Muris [96] suggested that the efficacy of these interventions could be improved by training adequate reactions in terms of shame and also guilt. This means that teaching children to regulate and reduce excessive self-conscious emotions could help to reduce externalizing and internalizing symptoms. Likewise, the authors of [97] underlined the importance for cognitive therapies and preventive interventions to target both the cognitive and affective components of children’s self-view. In this perspective, while feelings of sadness, loss of hope, irritability, and anxiety are commonly explored in clinical contexts, we suggest professionals to give relevance to secondary emotions too in an attempt to investigate more extensively the child’s emotional range: this means shame should be considered as a cross-diagnosis construct, with significant applicability in terms of clinical outcomes. For this reason, specific tools should be administered to children in order to monitor their levels and experiences of shame, while furnishing screening and preventing strategies: we believe this could help to guarantee better and more adaptive shame management.

Indeed, the relevant psychosocial aspect of shame should be carefully examined, especially in those children (and their parents) who self-refer to a child psychiatric service for emotional and/or behavioral problems. Thus, an integrated intervention is often recommended, and it should be aimed at investigating and treating psychological problems both in the child and in his/her caregivers, for example through an ecological evaluation of the whole family unit, as well as individual psychotherapy, family psychotherapy, and parental counselling. Moreover, before any medical or psychological intervention, parents and other caregivers should listen more carefully to the hidden needs of the youths, especially given that shame, more than other self-conscious emotions, could be very difficult to detect and recognize.

Practical implications of our findings can also apply to parents/caregivers. Family environment is fundamental in monitoring and determining children’s wellbeing, both in a positive and a negative fashion. Some of the studies here reported highlight how parents can interfere with shame’s management and contribute to its dysregulation. This gives us the urge to put all of our efforts into favoring parents’ awareness of the risks related to the excessive use of shame as a teaching style: in particular, professionals and educators should encourage an extensive use of parent/child dialogue, prompting the development of dialogical skills which can make it easier for children to understand why they are feeling that way, why they did wrong, and how they can learn and grow up from that experience. In particular, parents should be helped to focus on the importance of a good and honest communication with the child, especially when it comes to transgressions: more specifically, they should put massive attention into avoiding causing their children to feel defective as an individual, concentrating all of the blaming onto actions rather than on the child as a whole.

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
