# Peer review of "A Narrative Review about Prosocial and Antisocial Behavior in Childhood: The Relationship with Shame and Moral Development"

_children, 2022, doi:10.3390/children9101556_

Round 1

Reviewer 1 Report

The paper focuses on a very relevant topic, as theoretical discussion on the association of shame with adaptive/nonadaptive behavior is often nonconclusive and controversial. It also breaks down with the theoretical proposals which oppose shame to guilt, considering their adaptive and nonadaptive role, respectively. The assumption of this self-conscious emotion as dichotomous, leading both to prosocial and disruptive responses, contributes to conceptually frame shame in a more complex and comprehensive way. Furthermore, it proposes an extensive reflection of the effects of cultural differences and parenting styles on the experience of shame, as well as on the implications of the research on this emotion for promotional, preventive and remediation interventions.

Nevertheless, the publication of the manuscript is not feasible, as it presents severe theoretical and methodological weaknesses.

The aim/s and the focus of the review are not clear. Although antisocial and prosocial behavior are closely linked to moral development, they are not linearly overlapping constructs. Also, this work is mostly centered on the origins of shame and the association of this self-conscious emotion with children’s moral development, and not on these specific behavioral outcomes. In line with this, the initial paragraph could be clearer and more incisive, as the review’s focus is not on the neuropsychological development, or on the importance of the interaction of emotion and cognition on moral development, but, rather, on the effects of shame on moral behavior.

Along with these issues, as the focus of your review is prosocial and antisocial behaviors, why were these two terms not included in your search terms? The inclusion of these terms in your query may increase the number of articles focusing on the association of shame with moral development. Similarly, the inclusion of the expression ‘self-conscious emotions’, in line with Michael Lewis model (cf., Lewis, 1992, for additional information on the initial formulation of this model), which is widely cited, would possibly improve your search robustness. Additionally, the reference to ‘preadolescence’ should not be included, as it sounds a bit confusing. It would be clear to assume that only studies with children aged up to 12 years-old are to be considered.

The search terms are not coherent with the results analysis, as the cultural differences, parental influences, and the association of shame/moral emotions with psychopathology were not considered in your query. In addition, the title of the results’ subsection ‘Psychopathological Consequences of Primary Moral Emotion Dysregulation’ is not appropriate, as, in this subsection and along the literature review conducted, the focus is shame, and not other self-conscious emotions, although exploring the interaction of this emotion with the other moral emotions is relevant. Similarly, the authors report, not only the effects of self-conscious emotions on the development of socioemotional adjustment problems, as psychopathological markers, but also the association of shame with adaptive development, namely with prosocial behavior. This represents a conceptual problem. Please also note that the subsections of the results must be articulated in a more consistent way. For example, what link do the authors establish between cultural differences and parental influences in the development and expression of shame? And between these two dimensions and the psychopathological consequences of this emotion dysregulation?

Based on these significant issues of concern, I consider that the publication of the paper is not feasible.

Author Response

Thank you for your honest review, which could certainly help us improve our work. We modified the initial paragraph and the whole introduction as you suggested, trying to make it clearer and more incisive. As showed in the Materials and Methods section, we have now included the terms ‘prosocial behavior’ and ‘antisocial behavior’ in our search terms (please see page 3, paragraph 2.1, lines 4-7). The results of this new search added only 3 studies which we did not consider before, but unfortunately they did not meet our inclusion criteria and therefore they were excluded from our results. We thought about adding ‘self-consciuous emotions’ among the search terms of our query as well, but having been already very specific, most of the studies we retrieved were about both shame and guilt, and our aim was to consider shame alone in the process. Furthermore, we excluded the reference to preadolescence as suggested, please see page 3, paragraph 2.2, line 4. We thought a lot about the best terms to include in our search, but we structured the result section on the basis of the main topics of the retrieved studies, and those topics (parental influences, cultural differences, etc.) were obtained through the search, therefore after the analysis of the results. Furthermore, we modified the title of the last results’ subsection as suggested (please see page 7, paragraph 3.4). Finally, we tried to explain better all the limitations of our work in the dedicated paragraph of the discussion. We hope to have improved our work and made the paper clearer. 

Reviewer 2 Report

I have no comments other than 2 very small typos: On the opening page, at the beginning of Paragraph 3, there is a missing word:  "This makes it easier . . ."  Then on page 1, Paragraph 3, there is another missing word:  "So it's not a big surprise . . .  Otherwise, it was a clean article in this regard. 

Content-wise, I have no comments either as you developed a well-organized paper that addressed your key issues. The bibliographical list is strong and varied. While at first, I was worried about the age of the resources, once I began to see more contemporary sources, I relaxed. If you were to revisit this study, I would suggest that you include studies from the field of neuroscience and neuropsychology as more development is a theme that is being reviewed within that discipline and would provide interesting insights into behavior as it relates to shame and guilt. 

Author Response

Thank you for your review and its comments. We fixed the typos you noted. Moreover, we had the opportunity to add some data we retrieved from literature about the relationship between shame proneness and neurosciences. Please see page 7, paragraph 3.4, lines 7-16.

Reviewer 3 Report

The article is devoted to the extremely urgent problems of the consequences of the imposition of shame at the level of behavior. The authors have attracted quite a lot of research to work with their hypothesis, which has very serious prospects. It is important to note that a fairly extensive review allowed the authors to reach the level of theoretical generalization of the problem. At the end of the article, the authors outline a number of subsequent studies that will allow deeper insight into the psychology of the phenomenon under study.

Author Response

Thank you for your comments. We agree on the fact that this is an important and urgent topic and it needs further studies in order to clarify its scope.

Reviewer 4 Report

The manuscript presents an interesting review of the literature related to prosocial and antisocial behaviour in childhood. Specifically, the relationship of the behaviour with shame and moral development in children under 12 years of age. The inclusion and exclusion criteria are clearly identified, and an analysis is made of the common and differentiating elements of the articles analysed, with an appropriate structuring by area of interest. Its main limitation is that it is not empirical, nor does it perform a statistical analysis to support the results of the meta-analysis. This is probably due to the impossibility of having more detailed data or results for each of the studies analysed. The discussion provides valuable recommendations for the scientific community regarding future lines of research on shame.

One recommendation would be to give more emphasis in the results on age and gender. This information is included in the analysis of each study, but could be synthesised in separate sections.

Author Response

Many thanks for your review.  We tried to summarize a massive work, and our result might be just a first step in a more intensive effort about this topic. Unfortunately, the extreme heterogeneity of the included studies may have been detrimental to contain data about age and gender in a way we would have liked. Nevertheless, we tried to explain this concept better in the discussion section.

Round 2

Reviewer 1 Report

Dear Authors,

The revision performed substantially improved  the manuscript’s quality.

The first paragraph of the introduction is now more incisive and clearer, as it focuses on moral development and its impact on the individuals’ functioning. The research goals are now clearer and coherent with the way the results are presented, despite requiring a grammatical revision.

I only suggest the authors to revise the last sentence of the introduction, as, as far as I understood, the focus of your revision was not to analyze studies on the practical implications of the shame association with moral development and behavioral outcomes for child mental health professionals, educators, and parents, but instead to reflect on these implications based on the results observed. This must also be revised in the abstract.

In the Materials and Method’s section, the changes introduced both in the keywords used in the query and in the inclusion criteria (i.e., children’s age) made these methodological issues more robust.  

In the Results, the revision of the title of the 3.4. subsection made the articulation with the previous sections clearer and consistent.

In the discussion, the reflection on the study’s limitations, namely on the heterogeneity of the studies included in the revision, helps the reader to better understand the strategy used to describe the results.

Overall, in its current form, the scientific relevance and soundness of your work are now more visible.

Author Response

Thank you for your reply. We checked again the paragraph you mentioned and we revised the grammar. Moreover, we modified as suggested both the last sentence of the introduction and the corresponding part of the abstract. We hope the introduction is now clearer.